# Study on the Trend in Microbial Changes during the Fermentation of Black Tea and Its Effect on the Quality

**DOI:** 10.3390/foods12101944

**Published:** 2023-05-10

**Authors:** Changwei Liu, Haiyan Lin, Kuofei Wang, Zhixu Zhang, Jianan Huang, Zhonghua Liu

**Affiliations:** 1Key Laboratory of Tea Science of Ministry of Education, Hunan Agricultural University, Changsha 410128, China; 2National Research Center of Engineering Technology for Utilization of Functional Ingredients from Botanicals, Hunan Agricultural University, Changsha 410128, China; 3Co-Innovation Center of Education Ministry for Utilization of Botanical Functional Ingredients, Hunan Agricultural University, Changsha 410128, China; 4Key Laboratory for Evaluation and Utilization of Gene Resources of Horticultural Crops, Ministry of Agriculture and Rural Affairs of China, Hunan Agricultrual University, Changsha 410128, China

**Keywords:** black tea, fermentation, bacteria, fungi, quality chemistry

## Abstract

The role of tea endophytes in black tea fermentation and their impact on black tea quality remain unclear. We collected fresh leaves of Bixiangzao and Mingfeng tea and processed them into black tea, while testing the biochemical composition of both the fresh leaves and the black tea. We also used high-throughput techniques, such as 16S rRNA, to analyze the dynamic changes in the microbial community structure and function during black tea processing in order to investigate the influence of dominant microorganisms on the quality of black tea formation. Our results showed that bacteria, such as *Chryseobacterium* and *Sphingomonas*, and *Pleosporales* fungi dominated the entire black tea fermentation process. Predicted functional analysis of the bacterial community indicated that glycolysis-related enzymes, pyruvate dehydrogenase, and tricarboxylic acid cycle-related enzymes were significantly elevated during the fermentation stage. Amino acids, soluble sugars, and tea pigment content also increased considerably during fermentation. Pearson’s correlation analysis revealed that the relative bacterial abundance was closely related to the content of tea polyphenols and catechins. This study provides new insights into the changes in microbial communities during the fermentation of black tea and demonstrates understanding of the basic functional microorganisms involved in black tea processing.

## 1. Introduction

Tea is a globally popular beverage and black tea is the most produced and consumed type of tea, representing over 70% of the world’s tea production [1,2,3]. Black tea is appreciated by consumers for its smooth sweetness, clear soup color, floral and fruity aroma, and health benefits, such as antioxidant properties and regulation of blood lipids [4,5,6].

Tea tree endophytic bacteria play a crucial role in influencing the tea tree’s response to its growth environment, status, metabolites, and the flavor of its leaves, through their interactions with tea trees as they grow within their tissues and organs [7,8]. Although there is a long-term synergistic evolutionary relationship between endophytic bacteria and tea trees, research on the endophytic bacteria of tea trees started late and is still underexplored [9]. Currently, preliminary exploration of the species and distribution characteristics of endophytic bacteria in tea trees, their interactions with tea trees, and their biological functions has been carried out [10]. Fermentation is a critical step in the processing of black tea [11]. During the kneading process, mechanical forces and enzymes in tea cells catalyze the oxidation and polymerization of catechins, resulting in the formation of black tea’s flavor substances such as theaflavins, thearubigins, and theabrownin [12,13]. The oxidation of catechins is responsible for giving black tea its characteristic red soup and leaves, as well as its rich aroma and sweet taste [14]. During the fermentation process of black tea, a complex microbiota exists, with microbial populations performing a variety of functions including the transport and metabolism of amino acids, carbohydrates, nucleotides, and fatty acids, coenzyme transport and metabolism, and the biosynthesis, transport, and decomposition of secondary metabolites [15,16]. Roughly 50% of bacterial and fungal taxa present during tea fermentation are similar to the endophytic bacteria found in tea, with these shared taxa accounting for more than 90% and 80% of the total abundance of bacteria and fungi, respectively [17]. Through correlation analysis between the chemical quality of tea leaves and the architecture of endophytic bacterial populations in tea leaves, it has been confirmed that several endophytic bacterial taxa are highly significantly correlated with the chemical fraction content of tea leaves, suggesting their involvement in the synthesis of flavor substances in black tea [18].

In recent years, there has been growing interest in the role of microorganisms in the formation of tea, particularly in the quality of dark tea. Studies have identified various microorganisms, such as *Cyberlindnera*, *Aspergillus*, *Klebsiella*, and *Lactobacillus*, which are involved in the formation of dark tea quality characteristics [19,20]. The physicochemical conditions during fermentation create a favorable environment for the growth of microorganisms [21]. Although microorganisms play a crucial role in the formation of black tea flavor during fermentation, there is limited information available about the composition of microbial communities during black tea fermentation and their potential functions in black tea processing. Previous research has indicated that *Chlamomonas* increases in abundance after wilting, kneading, and fermentation, compared to the initial stage, whereas *Methylobacteria* and *Rhizobia* are the dominant flora during black tea processing [17,22]. Therefore, it can be inferred that microorganisms play a crucial role in the formation of black tea quality. To explore this further, we conducted a study to analyze the dynamic changes in the composition, structure, and function of microbial communities during black tea fermentation, as well as the important quality-related metabolites of black tea. Additionally, we evaluated the correlation between the microbial communities and quality-related compounds to reveal the role of microbial communities in the formation of black tea quality. The goal of this study was to provide a theoretical foundation for the distinctive flavor and quality of black tea.

## 2. Materials and Methods

### 2.1. Experimental Materials

In the experiment, we selected 15 healthy tea plants each, of Bixiangzao (BXZ) and Minfeng (MF), that were uniform in growth and free from diseases and pests. From each plant, we collected one bud and three leaves from their fully expanded leaves. We divided the samples into two parts: one part was freeze-dried, crushed, and analyzed for the chemical composition of tea quality, while the other part was aseptically treated and subjected to nucleic acid extraction following the method of Lin et al. [23]. We also collected 500 g of tea leaves and processed them aseptically to make black tea using traditional methods. The tools used for black tea production were autoclaved, and fermentation was carried out in a sterile wire basket. We collected tea samples after the fermentation was completed and divided them into two parts: one part was freeze-dried and crushed for chemical composition analysis of tea quality; the other part was used for microbial genomic DNA extraction. The experimental samples were named Bixiangzao fresh leaves (BZX), Bixiangzao fermented leaves (BZF), Minfeng fresh leaves (MFX), and Minfeng fermented leaves (MFF).

Tea leaves were sterilized by referring to the method of Lin [23] et al., i.e., tea leaves were washed with sterile water for 30 s, 70% sterile ethanol for 2 min, 2.5% NaClO (containing 0.1% Tween 80) for 5 min, and then transferred to 70% sterile ethanol for 30 s. Finally, plant tissues were washed three times using sterile water, and were considered sterile.

### 2.2. Reagents and Instruments

The vacuum freeze dryer (Christ-Alpha, Osterode, Germany), ultrasonic cleaner (DTC-27, Suzhou City, China), ultrapure water preparation machine (Millipore-Q, Bedford, MA, USA), centrifuge (Hettich Rotina-380, Tuttlingen, Germany), Nanodrop (Thermo-2000, Waltham, MA, USA), gel imaging system (Thermo-Gene, Waltham, MA, USA), 0.2 μm filter membrane, sand core funnel, shaker, centrifuge tube and other instrument consumables were all domestic. Other instruments included a high-performance liquid chromatograph (LC2020, Shimadzu, Kyoto, Japan) and chromatographic column (Shimadzu, Kyoto, Japan, SB-C18, 4.6 × 250 mm, 5 μm).

An agarose gel recovery kit, PCR mix, markers, sterile enzyme-free water, PBS solution, ethanol, sodium hypochlorite, Tween 80, and liquid nitrogen were also used.

### 2.3. Quality Chemical Composition Assay

The chemical composition of tea quality was analyzed according to the method described by Jia et al. [24]. The content of tea polyphenols was determined using the forintanol method, whereas the content of six catechin components, three alkaloids, amino acid components, and four sugar components was measured by high performance liquid chromatograph (HPLC). The tea pigments, including theaflavins, thearubigins, and theabrownin, were determined using a systematic analysis method.

To prepare the tea broth, 3 g of crushed tea sample was weighed and placed in a 500 mL conical flask. Then, 450 mL of boiling water was added, and the mixture was subjected to a 100 °C water bath for 45 min, stirring every 10 min. After filtration and washing the residue, the filtrate was combined in a 500 mL volumetric flask, cooled, and fixed. The mixture was shaken well before analysis. Samples for HPLC analysis were passed through a 0.45 μm microporous membrane before detection.

### 2.4. Tea Tree Endophytic Bacteria Collection and Total DNA Extraction

The microbial assay was performed by referring to the method of Lin et al. [23]. The microbial DNA obtained from tea leaves was extracted and PCR amplified using primers 1114F/1392R. After completion of PCR, the PCR products were recovered by cutting the gel after passing the electrophoresis test. After measuring the quality and concentration of PCR products, the PCR products of different samples were mixed equally and then entrusted to Shanghai Meiji Biomedical Co. (Shanghai, China). All experiments were performed in 3 replicates.

### 2.5. Structural Analysis of Microbial Population Composition

The sequencing results were processed using FLASH and Trimmomatic to obtain valid data, which were then spliced into the QIIME platform. Non-redundant sequences were clustered into operational taxonomic units (OTUs) at a 97% similarity threshold using usearch, and the resulting representative sequences were used for taxonomic analysis. The RDP classifier Bayesian algorithm was used to assign taxonomic classifications to each OUT at each level, including kingdom, phylum, class, order, family, genus, and species. The community composition was determined at each taxonomic level, and comparisons were made using the Greengene database (Release 13.5) for bacterial and fungal analysis. Additionally, PICRUSt was used to normalize the OUT abundance tables, and information on COG families and KEGG Orthologs (KO) was obtained. The abundance of each COG and KO was calculated.

### 2.6. Data Analysis

All experiments were set up with more than three groups of biological replicates, and the data were expressed as mean ± SD, and were analyzed and processed using SPSS 28.0 statistical software. The comparison between the two groups of data was conducted by t-test, and the comparison of multiple groups of data was analyzed by one-way ANOVA. The graphics were generated by the Majorbio cloud platform (https://cloud.majorbio.com/ (accessed on 13 May 2021). The quantitative results are expressed as mean ± SEM. All experiments were performed in 3 replicates. *, *p* < 0.05; **, *p* < 0.01; ***, *p* < 0.001. *p* < 0.05 was considered significant.

## 3. Results

### 3.1. Sequence Statistics of Tea Bacteria and Fungi

High-throughput sequencing of 16S rRNA revealed that a total of 249,334 valid sequences were obtained from a total of 12 samples of fresh and fermented leaves from two varieties of tea tree, with an average of 20,777 sequences per sample. A total of 112 endophytic bacterial OTUs were obtained after dividing OTUs with 97% sequence similarity (Figure 1A). A total of 5 bacterial phylum, 11 bacterial classes, 27 bacterial orders, 46 bacterial families, and 77 bacterial genera were obtained by comparison analysis with the bacterial database Greengene.

High-throughput sequencing of the ITS region yielded a total of 671,132 valid sequences from 12 samples of fresh and fermented leaves from two varieties of tea tree, with an average of 55,927 sequences per sample. After clustering OTUs with 97% sequence similarity, 145 endophytic fungal OTUs were identified (Figure 1E). Comparison analysis with the fungal database Greengene revealed the presence of 3 fungal phyla, 13 fungal classes, 27 fungal orders, 50 fungal families, and 69 fungal genera.

Alpha diversity analysis, which encompasses species richness and evenness, is commonly used to assess microbial community ecology species diversity and can be used to demonstrate tea microbial species diversity [25]. Both microbial Shannon and Simpson’s indices increased in response to fermentation in both tea samples, indicating that fermentation resulted in changes in both tea microbial diversity and richness (Figure 1). Principal component analysis (PCA) of the amplicon sequence variants (ASVs) of tea microorganisms at the phylum level revealed that the bacterial populations in each group exhibited some differences, with PC1 and PC2 accounting for 69.23% and 17.18% of the variance, respectively, and cumulatively containing 86.41% of the ASV data at the phylum level. Similarly, the fungal populations in each group exhibited some differences, with PC1 and PC2 accounting for 59.22% and 16.48% of the variance, respectively, and cumulatively containing 75.7% of the ASV data at the phylum level. Notably, the fresh and fermented leaves of BXZ were clearly separated (Figure 1D,H).

### 3.2. Comparison of Tea Microbial Communities and Abundance during Black Tea Fermentation

During the fermentation of black tea, microorganisms play an important role in the formation of black tea flavor quality. By high-throughput sequencing of 16S rRNA, 135,260 valid sequences were obtained from tea samples of two varieties during fermentation, with an average of 22,543 sequences per sample. A total of 150 endophytic bacterial OTUs were obtained after dividing OTUs with 97% sequence similarity. Among them, 98 and 119 endophytic bacterial OTUs were detected in the fermented tea leaves of BXZ and MF, respectively. Taxonomic analysis revealed that the major bacterial taxa in the fermentation process of both varieties of tea were similar, with the *Chryseobacterium* bacterial genus being the dominant taxon, accounting for 50.29% and 35.93% of the bacterial populations in the fermented tea leaves of BXZ and MF, respectively. The *Sphingomonas, Acidovorax, Microbacteria,* and *Rhizobium* bacterial genera followed, on average accounting for about 12.34%, 8.74%, 6.28%, and 7.38% of the bacterial populations in BXZ fermented tea leaves, respectively, and about 16.34%, 7.95%, 9.07%, and 5.24% in MF fermented tea leaves (Figure 2A). Although the major bacterial taxa in the fermentation process of the two varieties of tea were similar, their relative abundances were somewhat different. Moreover, some low-abundance taxa, such as *Herbaspirillum*, *Devosia*, and *Morganella*, showed significantly different distribution in the two fermented teas.

To investigate the contribution of fresh leaf endophytic bacteria to the bacterial populations involved in the tea fermentation process, we analyzed the homology between the endophytic bacteria found in tea trees and those present during the fermentation process. Our analysis revealed that 59 and 65 bacterial taxa were shared by tea leaves and tea endophytic bacteria during the fermentation process of BXZ and MF, respectively, accounting for approximately 50% of the total number of bacterial populations involved in the fermentation processes. This suggests that about half of the bacterial taxa involved in the microbial fermentation process of tea leaves might have originated from tea tree endophytic bacteria. Interestingly, these shared bacterial taxa were also found to be the dominant ones, accounting for more than 95% of the total bacterial abundance in the fermentation process (Table 1). Our co-occurrence network analysis further supported this finding, as it revealed that the majority of the dominant bacterial populations in the fermentation process originated from the leaf endophytes (Figure 3A). Taxonomic analysis identified three bacterial phyla (*Proteobacteria, Bacteroidetes*, and *Actinobacteria*) and 16 bacterial genera (including *Chryseobacterium*, *Microbacterium*, and *Acidovorax*) that belonged to *Flavobacteria, Actinobacteria*, *Alphaproteobacteria*, *Betaproteobacteria*, *Gammaproteobacteria*, *Flavobacteriales*, *Micrococcales*, *Burkholderiales*, *Microbacteriaceae,* and other 12 bacterial families.

The changes in bacterial flora during fermentation also varied between tea varieties. Fermentation in the BXZ tea sample led to an increase in bacteria such as *Chryseobacterium*, whereas fermentation in the MF tea sample led to a decrease in these bacteria. This may be related to different inclusions in the different tea trees. *Devosia* and *Methylobacterium* decreased in both teas, whereas *Rhizobium*, *Sphingomonas*, *Microbacterium*, etc. increased in both teas.

### 3.3. Comparison of Endophytic Fungal Communities and Abundance in Tea during Processing

Through ITS high-throughput sequencing, a total of 342,691 valid sequences were obtained from tea samples of two varieties, 12 during fermentation, with an average of 57,116 sequences per sample. A total of 342,691 valid sequences were obtained, with an average of 57,116 sequences per sample. We identified 293 endophytic fungal OTUs with 97% sequence similarity, including 254 endophytic bacterial OTUs during the fermentation of BXZ tea leaves, which was significantly higher than that of MF tea leaves (121). Taxonomic analysis revealed that *Pleosporales* was the dominant endophytic fungal group during the fermentation of both varieties of tea, accounting for 53.25% and 77.15% of the fungal populations in fermented BXZ and MF tea leaves, respectively (Figure 2B). *Euotiales* and *Capnodiales* were also found to be present in both varieties, accounting for approximately 16.84% and 6.91% of the fungal populations in fermented BXZ tea leaves on average, and about 5.43% and 1.93% of the fungal populations in MF tea leaves. Although the main fungal taxa during fermentation were similar, their relative abundances differed significantly between the two varieties. Moreover, some fungal orders were only found in low abundance during the fermentation of BXZ tea leaves, including *Trichosporonales*, *Ustilaginales*, *Microascales*, *Magnaporthales*, *Taphrinales*, *Tilletiales*, *Chaetosphaeriales*, *Diaporthales*, *Agaricostilbales*, and *Auriculariales*, whereas *Myriangiales* was exclusively found during the fermentation process of MF tea leaves.

To better understand the role of fresh leaf endophytes in the formation of fungal populations during the fermentation process, we conducted a comparative analysis of tea endophytes and fungal populations. Our findings showed that, during the fermentation process of BXZ and MF, tea leaves and tea endophytic fungi shared 141 and 91 fungal taxa (OTUs), respectively, which accounted for approximately 50% of the total tea fungal populations. This suggests that roughly half of the fungal taxa involved in the microbial fermentation process of tea leaves were derived from tea endophytic fungi. However, despite their lower numbers, these taxa represented over 80% of the total fungal abundance in the fermentation process (as shown in Table 2), indicating that they were the primary fungal populations involved in the process. Co-occurrence network analysis further revealed that most of the dominant fungal populations during fermentation were derived from leaf endophytes (as shown in Figure 3), with the majority belonging to *Pleosporales* (order Gymnosporales) and *Capnodiales* fungi. Notably, the abundance of fungal species changed significantly during tea fermentation, with an increase in fungi such as *Capnodiales* and *Pleosporales*. Among these, the Chaetothyriales fungus was found to be specific to BXZ tea trees.

### 3.4. Functional Prediction Analysis of Endophytic Bacteria and Bacterial Populations during Fermentation in Fresh Tea Tree Leaves

The functions of the 16S were predicted using PICRUSt. Functional information was compared to the COG library, which revealed that the COG functional composition of all samples was relatively similar compared to the species composition (Figure 4). Both fresh leaf endophytes and bacterial populations during fermentation played a role in various metabolic functions including, but not limited to, energy production and conversion, transport and metabolism of amino acids, carbohydrates, nucleotides, and fatty acids, transport and metabolism of coenzymes, biosynthesis, transport and catabolism of secondary metabolites, and transport and metabolism of inorganic ions, as well as biodefense and control.

The analysis of KEGG pathway abundance and composition (at level 3) of endophytic bacterial populations in tea trees revealed that these bacteria mainly enriched pathways related to the metabolism and biosynthesis of amino acids (such as alanine, aspartic acid, glutamic acid, tyrosine, tryptophan), as well as the metabolism of sugars and lipids. These results indicate that endophytic bacterial metabolic functions within tea tree tissues are diverse, and differences in the composition, structure, and activity of these populations may directly affect tea quality. However, there were notable differences in the abundance of bacteria involved in metabolism during the fermentation of fresh leaves, with the highest abundance occurring during fermentation and the second highest in fresh leaves.

### 3.5. Comparison of the Content of Main Quality Chemical Components of Tea Leaves in Different Varieties of Fresh Leaves and in the Fermentation Process

The taste of tea soup is mainly influenced by L-theanine and other amino acids present in tea leaves, which are important flavoring substances and are closely linked to tea quality [26,27]. This study’s findings reveal that MF had significantly higher levels of both theanine and total free amino acids when compared to BXZ (Table 3). The total free amino acid content in MF was 2.96 ± 0.006%, whereas in BXZ, it was 2.69 ± 0.002%, and the difference was statistically significant (*p* < 0.05). A comparative analysis of amino acid fractions in BXZ and MF revealed that the amino acid contents significantly increased after fermentation. The total free amino acid content in the fermented tea leaves was 3.18 ± 0.002% and 3.24 ± 0.004% in BXZ and MF, respectively, which is 0.28~0.49% higher than the content of fresh leaves before fermentation, and the differences were statistically significant (*p* < 0.05). However, the theanine content decreased significantly after fermentation, with BXZ decreasing from 0.51% to 0.44% and MF decreasing from 0.64% to 0.56%, and the differences were statistically significant (*p* < 0.05) when compared to the fresh leaves before fermentation.

Table 3 shows the significant differences in caffeine content between BXZ and MF, which were 2.215 ± 0.002% and 2.694 ± 0.003%, respectively, with theobromine being the main constituent and there being no detectable theophylline. The contents of gallic acid and six catechin fractions were significantly different between BXZ and MF, with DL-catechin (DL-C), epicatechin (EC), and epigallocatechin (EGC) being non-ester type catechins. The differences in contents between the two tea trees were highly significant. Gallocatechin gallate (GCG) content in BXZ and MF was 1.756 ± 0.002% and 2.378 ± 0.004%, respectively, with no significant difference (*p* > 0.05) in the total tea polyphenols, which were 17.588 ± 0.169% and 17.749 ± 0.157%, respectively. However, the theobromine, caffeine, and gallic acid contents in BXZ and MF old leaves were significantly reduced compared to fresh leaves, with theobromine decreasing the most, from 0.132 ± 0.001% and 0.165 ± 0.001% to 0.032 ± 0.001% and 0.957 ± 0.002%, respectively, showing highly significant differences (*p* < 0.01). The content of tea polyphenols and catechin components were also significantly reduced, with a decrease in each catechin component in BXZ ranging from about 7.06% to 19.01% and in MF ranging from about 22.24% to 37.09%, with significant differences (*p* < 0.05). After fermentation, the theobromine, caffeine, and gallic acid contents in BXZ and MF tea leaves significantly increased, with significant differences (*p* < 0.05) from 0.132 ± 0.001% and 0.165 ± 0.001% to 0.153 ± 0.001% and 0.183 ± 0.002% in theobromine content, respectively, and from 2.215 ± 0.002% and 2.694 ± 0.002% to 3.667 ± 0.005% and 3.861 ± 0.016%, respectively, with significant differences (*p* < 0.05). The gallic acid content also increased from 0.06 ± 0.001% and 0.062 ± 0.002% to 0.46 ± 0.001% and 0.457 ± 0.001%, respectively, with significant differences (*p* < 0.05). However, the contents of tea polyphenols and catechin fractions were significantly reduced, especially DL-C, EC, and GCG, which were almost undetectable after fermentation.

The fructose and maltose contents of MF fresh leaves were significantly higher than that of BXZ, whereas their sucrose content was significantly lower than that of BXZ (*p* < 0.05). However, there was no significant difference in glucose content between the two tea varieties (*p* > 0.05). Compared to fresh leaves before fermentation, the content of fructose and maltose in tea leaves after fermentation increased significantly, from 0.15 ± 0.0001% and 0.171 ± 0.0001% to 0.59 ± 0.002% and 0.605 ± 0.002%, respectively, with significant differences (*p* < 0.05). Maltose content also increased significantly, from 1.271 ± 0.0005% and 1.534 ± 0.0002% to 1.748 ± 0.002% and 1.838 ± 0.005%, respectively (*p* < 0.05). Conversely, glucose and sucrose contents were significantly reduced, with glucose content decreasing from 0.857 ± 0.0001% and 0.853 ± 0.0001% to 0.561 ± 0.003% and 0.569 ± 0.001%, respectively, and sucrose content decreasing from 2.182 ± 0.003% and 1.899 ± 0.004% to 0.488 ± 0.001% and 0.581 ± 0.002%, respectively (*p* < 0.05). Theaflavins, thearubigins, and theabrownin are formed by the oxidative polymerization of catechins, catalyzed by enzymes in tea. No black tea trisulfide was detected in fresh tea leaves. After fermentation, theaflavins, thearubigins, and theabrownin increased significantly, with the three elements of black tea in MF being higher than those in BXZ.

### 3.6. Relationship between Dominant Microorganisms and Tea Quality

Figure 5 illustrates the correlation between the fresh leaves of the two tea varieties with regard to the chemical fractions of tea leaves during fermentation and the abundance of their dominant bacterial populations. The study found a significant correlation between the chemical quality of tea leaves and the abundance of some endophytic bacteria. The abundance of bacterial genera, such as *Devosia, Sphingomonas, Chryseobacterium, Microbacterium, Acidovorax, Rhizobium, and Herbaspirillum,* showed a highly significant positive correlation (*p* < 0.01), indicating their potential involvement in the chemical synthesis of tea chemicals. However, these bacterial populations were highly significantly negatively correlated (*p* < 0.01) with sucrose and glucose content, implying their involvement in sugar metabolism. Conversely, bacterial genera such as *Enterococcus, Raistonia, Bradyrhizobium*, and *Paraburkholderia*, were significantly negatively correlated with the content of total amino acids, theanine, maltose, gallic acid, theobromine, and caffeine in tea, but highly significantly positively correlated with sucrose and glucose content (*p* < 0.01). Methylobacterium was significantly and positively correlated with the content of tea polyphenols and catechin fractions EGC, EC, EGCG, ECG, etc. (*p* < 0.05). However, a number of bacterial genera, such as *Streptococcus* and *Rothia*, showed a significant negative correlation (*p* < 0.05) with the content of these fractions. These results suggest that some endophytic bacterial taxa may be involved in the biosynthesis of tea chemicals, whereas some are related to the transformation and metabolism of tea substances.

Figure 6 displays the correlation between the chemical fractions of tea leaves during fermentation, the fresh leaves of the two tea varieties, and their dominant fungal populations. The results indicate a significant correlation between the number of some endophytic fungi and the chemical quality of tea leaves. The fungal order *Pleosporales* showed a highly significant positive correlation (*p* < 0.01) with the content of total amino acids, theanine, maltose, gallic acid, theobromine, and caffeine in tea leaves. In contrast, the fungal orders *Sordariomycetes, Microascales, Xylariales, Rhizophydiales, Erythrobasidiales,* and *Chaetothyriales* were negatively correlated with these chemical fractions. The content of tea polyphenols and catechin fractions EGC, EC, epigallocatechin gallate (EGCG), and ECG had a positive correlation with the number of *Erythrobasidiales* and *Chaetothyriales*, but a negative correlation with the number of *Eurotiales* and *Malasseziales*. The content of glucose and sucrose in tea leaves was significantly and positively correlated with the content of fungal orders such as *Microascales, Xylariales, Chaetesphaeriales, Rhyzophydiales*, and others.

## 4. Discussion

During the processing of black tea, enzymatic systems are utilized in the withering, kneading, and fermentation stages [28,29]. Although black tea fermentation was once believed to not involve microorganisms, recent evidence has supported their involvement [30]. Our study found that a diverse microbiota of bacteria and fungi exist during black tea fermentation, and functional predictions confirmed their roles in amino acid, carbohydrate, nucleotide, and fatty acid transport and metabolism, coenzyme transport and metabolism, and secondary metabolite biosynthesis, transport, and catabolism. This suggests that microorganisms are likely to play a role in the fermentation process, despite its short duration. Furthermore, we found that about 50% of the bacterial and fungal taxa involved in the microbial fermentation process of tea were similar to tea endophytes, and that the relative abundance of these taxa accounted for more than 90% and 80% of the total abundance of bacteria and fungi in the fermentation process, respectively. This suggests that the majority of the dominant microbial populations in the fermentation process likely originate from leaf endophytes. Previous studies have identified Bacteroidetes, Actinobacteria, and Cyanobacteria as the main bacteria during black tea fermentation [23], whereas *Methylobacterium, Sphingomonas, Aureimonas,* and *Devosia* have been shown to facilitate the formation of black tea flavor substances [31]. Moreover, the genera *Sphingomonas*, *Chryseobacterium*, and *Aureimonas* have been correlated with kaempferol, theaflavins, thearubigins, and theabrownins. Although there are few studies on the biotransformation of tea quality components by endophytic bacteria, some research has suggested that microorganisms secrete polyphenol oxidase and peroxidase during fermentation to promote the biotransformation of tea components [32], which forms the basis for the formation of black tea quality, flavor, and physiological functions. Additionally, there have been preliminary explorations of the biotransformation of tea polyphenols, such as a study that found the use of fungal-derived laccase and fresh tea-derived polyphenol oxidase can catalyze the conversion of catechins into complex polyphenolic compounds [33].

We observed a significant positive correlation between the content of certain tea components and the number of some endophytic bacteria or fungi, including the bacteria *Devosia, Sphingomonas, Chryseobacterium, Microbacterium, Acidovorax, Rhizobium, Herbaspirillum* and the fungus *Pleosporales*. Conversely, the numbers of some endophytic bacteria or fungi, such as the bacteria *Enterococcus, Raistonia, Bradyrhizobium, Paraburkholderia* and the fungi *Sordariomycetes, Microascales, Xylariales*, and *Rhizophydiales*, were significantly negatively correlated with the content of tea quality components. Some endophytic bacteria are capable of producing aromatic substances, particularly during the fermentation process of black tea, whether these are endophytic or surface microorganisms, and how different factors regulate their activity requires further investigation. Additionally, the role of harmful bacteria and the effect of storage time on the abundance of beneficial bacteria in tea also warrant further research.

## Figures and Tables

**Figure 1 foods-12-01944-f001:**
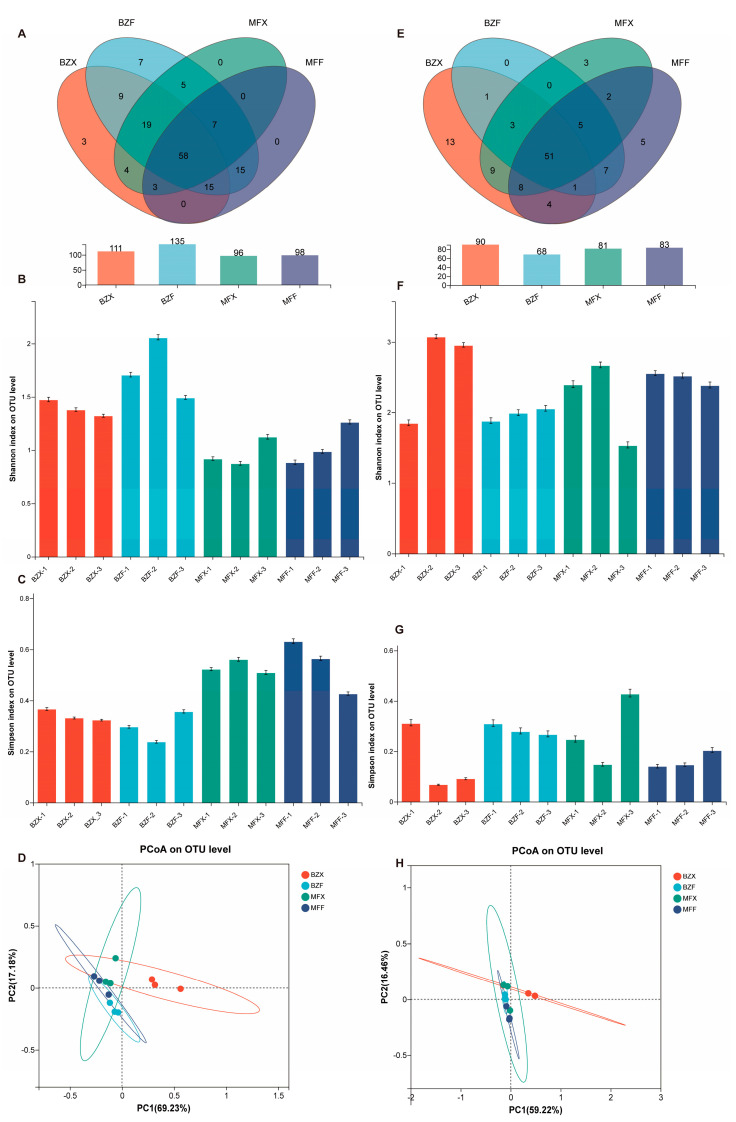
Tea microorganisms in each group α diversity and β diversity analysis. (**A**) Venn diagram of the number of endophytic bacterial OTUs in each group. (**B**,**C**) Shannon index and Simpson’s index of tea bacteria. (**D**) PcoA analysis of tea fungus. (**E**) Venn diagram of the number of endophytic fungal OTUs in each group. (**F**,**G**) Shannon index and Simpson’s index of tea fungus. (**H**) PcoA analysis of tea fungus.

**Figure 2 foods-12-01944-f002:**
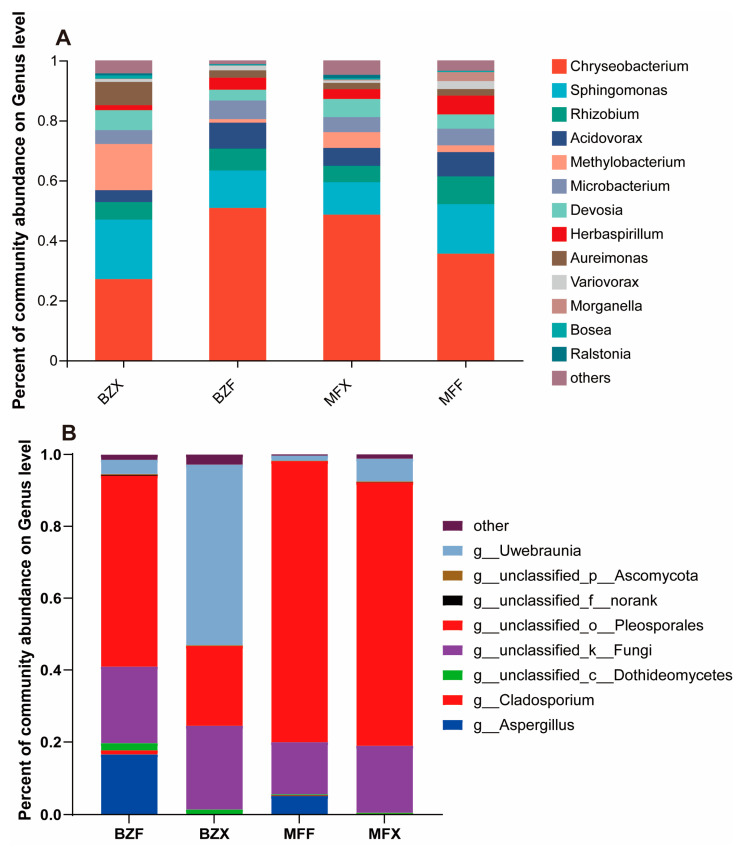
Genus-level microbial composition of tea group: (**A**) bacterial populations, (**B**) fungal populations.

**Figure 3 foods-12-01944-f003:**
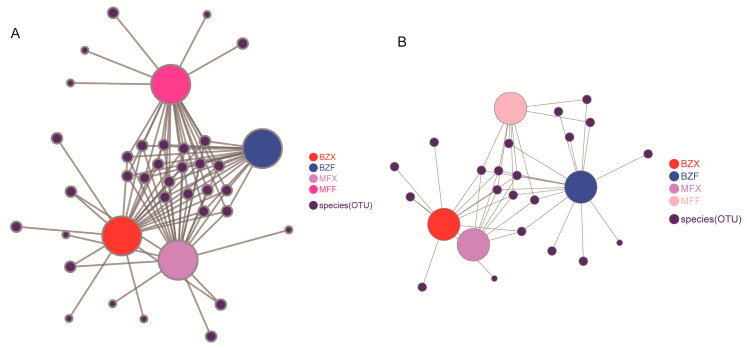
Correlation of tea microorganisms in each group: (**A**) bacterial populations, (**B**) fungal populations.

**Figure 4 foods-12-01944-f004:**
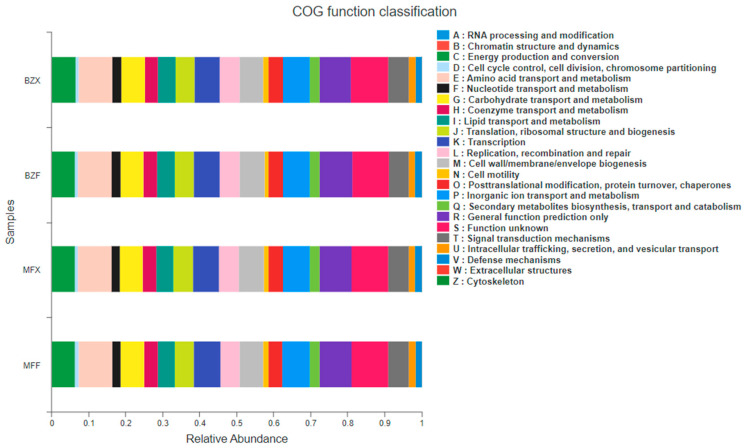
Prediction of the functions of bacterial communities in new leaves and fermented leaves using PICRUSt.

**Figure 5 foods-12-01944-f005:**
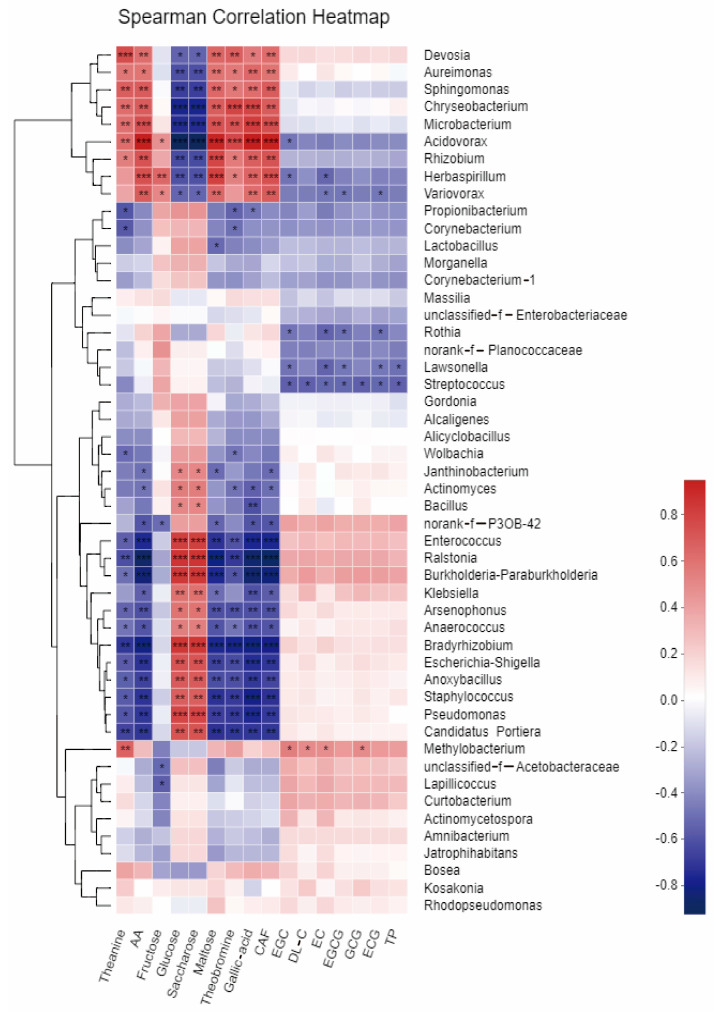
Heatmap analysis showing the correlation between the abundance of dominant bacterial species and chemical properties of tea leaves. *, *p* < 0.05; **, *p* < 0.01; ***, *p* < 0.001. *p* < 0.05 was considered significant.

**Figure 6 foods-12-01944-f006:**
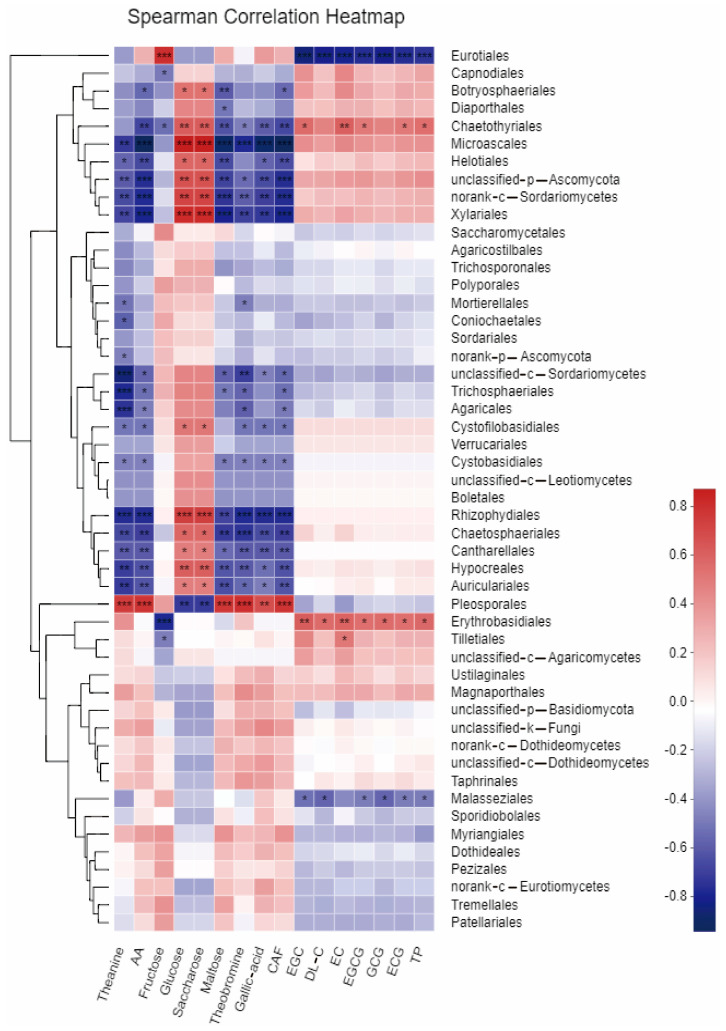
Heatmap analysis showing the correlation between the abundance of dominant fungal species and chemical properties of tea leaves. *, *p* < 0.05; **, *p* < 0.01; ***, *p* < 0.001. *p* < 0.05 was considered significant.

**Table 1 foods-12-01944-t001:** Dominant bacterial communities shared by endophytes in new leaves and bacteria detected in fermentation process.

Phylum	Class	Order	Family	Genus	BZF (%)	BZX (%)	MFF (%)	MFX (%)
*Actinobacteria*	*Actinobacteria*	*Micrococcales*	*Microbacteriaceae*	*Microbacterium*	6.28	4.60	5.23	4.89
*Bacteroidetes*	*Flavobacteriia*	*Flavobacteriales*	*Flavobacteriaceae*	*Chryseobacterium*	50.22	26.89	35.75	48.75
*Proteobacteria*	*Alphaproteobacteria*	*Rhizobiales*	*Aurantimonadaceae*	*Aureimonas*	2.50	5.21	2.18	2.07
			*Bradyrhizobiaceae*	*Bosea*	0.40	1.04	0.36	0.42
			*Hyphomicrobiaceae*	*Devosia*	3.46	6.84	4.92	6.19
			*Methylobacteriaceae*	*Methylobacterium*	0.58	4.56	0.79	0.94
					0.04	5.45	0.05	1.08
					0.29	3.06	0.77	0.97
					0.27	1.39	0.43	0.96
					0.08	0.65	0.13	0.62
			*Rhizobiaceae*	*Rhizobium*	6.72	4.09	8.15	2.29
					0.64	1.83	0.91	2.84
		*Sphingomonadales*	*Sphingomonadaceae*	*Sphingomonas*	6.53	4.93	7.47	4.11
					2.84	1.43	5.08	4.26
					0.38	9.13	0.50	0.63
					2.52	2.79	3.19	1.64
	*Betaproteobacteria*	*Burkholderiales*	*Comamonadaceae*	*Acidovorax*	8.74	3.94	7.95	5.91
				*Variovorax*	1.62	0.82	2.69	0.83
			*Oxalobacteraceae*	*Herbaspirillum*	2.15	1.59	3.57	1.54
					2.03	0.07	2.71	1.81
	*Gammaproteobacteria*	*Enterobacteriales*	*Enterobacteriaceae*	*Morganella*	0.03	0.04	2.34	0.26

Note: bacterial species (OTU) with relative abundance more than 0.5% are listed in this table.

**Table 2 foods-12-01944-t002:** Dominant fungal species shared by fungal endophytes in new leaves and fungal species detected in the fermentation process.

Phylum	Class	Order	BZF (%)	BZX (%)	MFF (%)	MFX (%)
*Ascomycota*	*Dothideomycetes*	*Capnodiales*	3.66	46.14	1.38	6.46
			0.24	3.19	0.11	1.78
			1.61	1.00	0.23	0.21
			0.99	0.06	0.09	0.04
		*Pleosporales*	44.89	22.34	67.79	54.96
			5.86	1.31	7.27	4.65
			2.07	0.48	1.94	12.49
			0.07	0.39	0.07	0.54
		*unclassified*	0.28	0.08	0.03	0.08
	*Eurotiomycetes*	*Chaetothyriales*	0.01	0.66	0.00	0.00
*Unclassified Fungi*	*unclassified*	*unclassified*	21.28	21.64	15.15	17.19

**Table 3 foods-12-01944-t003:** Contents of major chemical components in tea (%).

Sample	BZX	BZF	MFX	MFF
Theanine	0.51 ± 0.002	0.44 ± 0.002	0.64 ± 0.001	0.56 ± 0.001
Total amino acids	2.69 ± 0.002	3.18 ± 0.012	2.96 ± 0.005	3.24 ± 0.004
Theobromine	0.132 ± 0.001	0.153 ± 0.001	0.165 ± 0.001	0.183 ± 0.002
Theophylline	0	0	0	0
Caffeine	2.215 ± 0.002	3.667 ± 0.005	2.694 ± 0.003	3.861 ± 0.016
Gallic acid	0.06 ± 0.001	0.46 ± 0.001	0.062 ± 0.002	0.457 ± 0.001
D-LC	0.6 ± 0.001	0	0.725 ± 0.001	0
EC	0.559 ± 0.003	0	0.526 ± 0.001	0
EGC	3.63 ± 0.002	0.023 ± 0.001	3.435 ± 0.004	0.014 ± 0.001
EGCG	5.616 ± 0.006	0.13 ± 0.001	6.489 ± 0.008	0.1 ± 0.001
GCG	1.756 ± 0.002	0	2.378 ± 0.004	0
ECG	1.133 ± 0.001	0.11 ± 0.001	1.161 ± 0.001	0.056 ± 0.001
Total tea polyphenols	17.588 ± 0.169	9.351 ± 0.113	17.749 ± 0.157	7.610 ± 0.105
Phenol–ammonia ratio	16.285 ± 0.05	6.7611 ± 0.08	13.147 ± 0.08	5.708 ± 0.051
Fructose	0.15 ± 0.0001	0.59 ± 0.002	0.171 ± 0.0001	0.605 ± 0.002
Glucose	0.857 ± 0.0001	0.561 ± 0.003	0.853 ± 0.0001	0.569 ± 0.001
Sucrose	2.182 ± 0.003	0.488 ± 0.001	1.899 ± 0.004	0.581 ± 0.002
Malt dust	1.271 ± 0.0005	1.748 ± 0.002	1.534 ± 0.0002	1.838 ± 0.005
Theaflavins	0	0.281 ± 0.002	0	0.323 ± 0.001
Thearubigins	0	5.425 ± 0.002	0	5.948 ± 0.002

## Data Availability

The data presented in this study are available on request from the corresponding author. The data are not publicly available due to our research is not yet completed, and some data is not suitable for public disclosure.

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
