# Peer review of "Study on the Trend in Microbial Changes during the Fermentation of Black Tea and Its Effect on the Quality"

_foods, 2023, doi:10.3390/foods12101944_

Round 1

Reviewer 1 Report

Well detailed original work, with editing problems.

I have problems with mainly some wrong text wrapping and typing error.  See line 365, 374 and 378. Furthemore the Fig 5 due to its size, is unreadable and unappreciable. Line breaks are missing before headings. The title of Fig. 1 has moved to the next page... etc..

But the work has valuable infoirmation can help to understand the role of tea tree and leaves microbiota in different processing steps related sensory and other beneficial properties of fermented tea products. Authors clearly indicate further possible and necessary directions of the research.

1.What is the main question addressed by the research? To serve new data about the microflora of tea leaves, to determine the dominant groups, to investigate the development of microflora and its consequences during the fermentation and its consequences, especially with regard to endophytes. 2.Do you consider the topic original or relevant in the field? Does it address a specific gap in the field? The work provides a detailed, comprehensive picture of the natural microflora of the tea plant and its changes during fermentation. It also provides useful data on changes in polyphenols and colorants.
3.What does it add to the subject area compared with other published material? Detailed data related the fermentation of black tea leaves, considering the DNA identification of microorganisms, determining the most important groups or strains, and useful data about the polyphenols and pigments. 4.Are the conclusions consistent with the evidence and arguments presented and do they address the main question posed? The conclusions are consistent with the results and arguments presented. 5. Are the references appropriate? The references are adequate, but you could have used more defining sources, e.g. Works by Siti Nurmilah (2022), Amanda Gouveia Mizuta (2020) or Sravanthi Goud Burragoni & Junhyun Jeon (2020). 6.Please include any additional comments on the tables and figures. I have no further comments on the tables and figures. See the comment in the online rewiev.

Author Response

Thank you for your review. We have revised it as requested.

Reviewer 2 Report

The manuscript “Study on the trend of microbial changes during the fermentation of black tea and its effect on the quality” is a quite well-written study that had as its objective to study the presence of microorganisms thorough fermentation and also the biochemical composition of fresh leaves and the same leaves after fermentation. It is an interesting publication based on properly conducted research. It discusses research results on an important aspect, such as the knowledge of the main microorganisms present in black tea fermentation.

The manuscript has to be significantly improved by a native English speaker, and the introduction should place the study precisely in the current field of knowledge. Also, in the introduction, the authors could also indicate the limitations of the study

line 17 – it is not necessary to write Study with a superscript

lines 18 -23 – Please separate and rephrase the section, eventually, such as: “Fresh tea leaves and tea leaves after the fermentation was completed were collected simultaneously. The changes in black tea microorganisms were analyzed by high-throughput sequencing to observe the dynamic changes in microbial community composition, structure and function during the fermentation of black tea and to investigate the influence of dominant microorganisms on the intrinsic substances of black tea during the fermentation process.”

lines 36-37 – “Black tea is the most widely consumed region in the world”?

line 68 – please correct aspergillus to superscript

line 89 – please insert a space between “(BXZ)and” – there are other similar errors throughout the manuscript, please revise

line 90 – please specify the used freeze dryer – model, type and country of production – revise the whole manuscript for all instruments that were used

line 92 – please insert a space between the number and degree sign – revise the whole manuscript

section 2.3 – please specify at least a reference on which is based the used methodology – each method should be based on similar studies. (same for sections 2.4, 2.5)

lines 146 – 149 – this phrase should be rephrased, i.e. “PICRUSt normalized the OTU abundance tables to obtain information on the COG family corresponding to OTUs and KEGG Ortholog (KO) information. Each COG and KO abundance was calculated.”

line 158 – correct the last P value to lower script

lines 169, 171, 1n4 176 – please decide if otu is lower- or uppercase

line 417 – “some studies have shown” – please specify those studies

-          As there is a discussion section, there would be important to compare the obtained results with similar results from other studies.

Author Response

(The authors gave the same response as above.)

Reviewer 3 Report

Although I appreciate the effort put into the analytical part of this article, the author's ignorance of key terms (Amino acids can generate brown pyrazines with roasted aroma through the Merad reaction) are the absolute reason why the paper must be rejected. Due to this lack of knowledge of the subject, I am extremely suspicious of all the results of this work.

Author Response

Thank you for your review. Thank you for your review. We apologize for our mistake. We have revised the paper and hope to have an opportunity to review it again. thank you.

Round 2

Reviewer 2 Report

The authors considerably improved the article and implemented all the required corrections. Although with so many track changes, it is hard to precisely revise the manuscript.